# Chromosome-level assemblies of multiple *Arabidopsis* genomes reveal hotspots of rearrangements with altered evolutionary dynamics

Wen-Biao Jiao [1] & Korbinian Schneeberger [1,2]✉

Despite hundreds of sequenced *Arabidopsis* genomes, very little is known about the degree of genomic collinearity within single species, due to the low number of chromosome-level assemblies. Here, we report chromosome-level reference-quality assemblies of seven *Arabidopsis thaliana* accessions selected across its global range. Each genome reveals between 13–17 Mb rearranged, and 5–6 Mb non-reference sequences introducing copy-number changes in ~5000 genes, including ~1900 non-reference genes. Quantifying the collinearity between the genomes reveals ~350 euchromatic regions, where accession-specific tandem duplications destroy the collinearity between the genomes. These hotspots of rearrangements are characterized by reduced meiotic recombination in hybrids and genes implicated in biotic stress response. This suggests that hotspots of rearrangements undergo altered evolutionary dynamics, as compared to the rest of the genome, which are mostly based on the accumulation of new mutations and not on the recombination of existing variation, and thereby enable a quick response to the biotic stress.

---

[1] Max Planck Institute for Plant Breeding Research, Department of Chromosome Biology, Carl-von-Linné-Weg 10, 50829 Cologne, Germany. [2] Faculty of Biology, LMU Munich, Großhaderner Str. 2, 82152 Planegg-Martinsried, Germany. ✉email: schneeberger@mpipz.mpg.de

The individual genomes of sexually reproducing species are typically highly collinear to enable physical exchange of alleles during meiosis. This exchange ensures the generation of diversity and the removal of deleterious alleles[1] and at the same time protects the offspring from major mutations changing the karyotype of a genome[2]. Despite the obvious importance of preserving a common karyotype, the presence of genomic rearrangements suggests that the genomes are in fact not entirely collinear. Genomic rearrangements (and the resulting lack of allelic exchange) have been shown to contribute to population diversification including the evolution of different sexes[3] or life-history traits[4].

But even though the absence of collinearity can have drastic effects, there is hardly anything known about the actual degree of collinearity within populations as most of the current genome studies are not based on chromosome-level assemblies. The first complete assembly of a plant genome was the reference sequence of *A. thaliana* (Col-0), which was based on a minimal tiling path of BACs sequenced with Sanger technology[5]. Since then multiple hundred *Arabidopsis* genomes have been studied, however, most of these studies relied on short-read based resequencing or reference-guided assembly, where the identification of genomic rearrangements remained challenging[6–12]. In contrast, reference-independent, chromosome-level assemblies with almost complete reconstruction of the nucleotide sequence enable accurate identification of all sequence differences and would therefore reveal the degree of synteny across the genome[13]. So-far, however, there are only a few whole-genome de novo assemblies for *A. thaliana* available including a re-assembly of the reference accession Col-0 as well as assemblies of four different accessions including Cvi-0, KBS-Mac-74, L*er*, and Nd-1, which have been generated in different studies and have not been compared against each other[14–18].

Here we release chromosome-level assemblies of seven *Arabidopsis thaliana* accessions. We identify 13–17 Mb genomic rearrangements, 5–6 Mb non-reference sequence in each genome. We find genic copy-number variations in around 5000 genes, including ~1900 non-reference genes. We develop a metric called synteny diversity to quantify the collinearity between the genomes and identify 350 euchromatic hotspots of rearrangements regions where genome collinearity between the genomes are strongly impaired. Further evolutionary analysis suggests these regions are undergoing different evolutionary dynamics as compared to the rest of the genome, which contribute to the rapid response to biotic stress.

## Results

### Chromosome-level assemblies of seven *A. thaliana* genomes.
Using deep PacBio (45–71×) and Illumina (56–78×) whole-genome shotgun sequencing, we assembled the genomes of seven accessions from geographically diverse regions including An-1 (Antwerpen, Belgium), C24 (Coimbra, Portugal), Cvi-0 (Cape Verde Islands), Eri-1 (Eringsboda, Sweden), Kyo (Kyoto, Japan), L*er* (Gorzów Wielkopolski, Poland) and Sha (Shahdara, Tadjikistan) (Supplementary Table 1) (see Methods). The assembly of L*er* was already described in the context of the development of a whole-genome comparison tool used in this study[13]. The seven accessions (together with the reference accession Col-0) were initially used as the founder lines of *Arabidopsis* Multi-parent Recombination Inbreeding Lines (AMPRIL)[19] population and were selected to maximize the genetic diversity in this set. The contig assemblies featured N50 values from 4.8 to 11.2 Mb and were thus similar to other long-read assemblies of *A. thaliana* genomes. Chromosome-normalized L50 (CL50)[20] values were 1 or 2 indicating that nearly all chromosomes were assembled into

a few contigs only (Fig. 1, Table 1 and Supplementary Table 2). In comparison with the reference sequence, we found less collapsed repeat regions in each of the assemblies as well as 41 (out of 70) reference sequence gaps, which could be bridged with contigs of the other assemblies, suggesting that the reference sequence could be improved using long-read assembly (Supplementary Table 3).

We arranged 43–73 contigs of each assembly to chromosome-level pseudomolecules based on homology to the reference sequence. Even though these assemblies do rely on the reference sequence, we would like to point out that the sequence assembly itself was independent of the reference sequence, and that the contigs were large in general, implying that it is unlikely that we misplaced any of the contigs. To confirm this, we compared two of the chromosome-level assemblies with three different genetic maps, where we did not find even a single misplaced contig (Supplementary Table 4). The seven chromosome-level assemblies reached a total length of 117.7–118.8 Mb, which is very similar to the 119.1 Mb of the reference sequence (Table 1) and even included parts of the highly complex regions of centromeres, telomeres and rDNA clusters (Supplementary Data 1 and Supplementary Table 5). The remaining unanchored contigs had a total length of 1.5–3.3 Mb and consisted almost entirely of repeats. This agrees with gaps between the contigs, which were mostly introduced due to repetitive regions (Supplementary Table 6). Overall, we annotated 27,098–27,574 protein-coding genes in each of the assemblies, which is similar to the 27,445 genes annotated in the reference sequence[21] (Table 1, Supplementary Data 2 and Supplementary Tables 7–8) (see Methods).

### Identification of syntenic and rearranged regions.
By comparing each of the new assemblies against the reference sequence using the whole-genome comparison tool SyRI (V1.1)[13], we found 102.2–106.6 Mb of collinear regions and 12.6–17.0 Mb of rearranged regions in each of the genomes (Fig. 2a). The rearrangements included 1.5–4.2 Mb (33–46) inversions, 1.8–2.9 Mb (729–1192) translocations, and—most abundantly—polymorphic duplications, which comprised 7.2–8.7 Mb (4288–5150) within each of the individual genomes (Supplementary Table 9). Similar to small-scale sequence variation[22], rearrangements were not evenly distributed along the chromosomes, but were enriched in pericentromeres (Supplementary Table 10). Their lengths ranged from a few dozen bp to hundreds of kb and even Mb scale (Fig. 2b), including a 2.48 Mb inversion specific to chromosome 3 of Sha (Supplementary Fig. 1 and Supplementary Table 11), which explains the suppression of meiotic recombination in this region in hybrids including the Sha haplotype[23–25]. Sequence divergence in rearranged regions was generally higher as compared to collinear regions mostly due to an excess of local copy-gain and copy-loss variation in rearranged regions (Fig. 2a, Supplementary Fig. 2 and Supplementary Table 12).

### Gene copy-number variations and pan-genome.
Genomic rearrangements have the potential to delete, create or duplicate genes resulting in gene copy number variation (CNV). Based on the clustering of orthologous genes across all eight accessions[26] we found 22,040 gene families with conserved copy number, while 4957 gene families showed differences in gene copy numbers in at least one accession (Fig. 2c and Supplementary Table 13). Almost 99% of these copy-variable gene families had a maximum copy number of 5 or less, while only less than 10% of them showed more than two different copy numbers across the eight accessions (Supplementary Fig. 3). Among the copy-variable genes we found 1941 non-reference gene families including 891 gene families present in at least two of the other accessions (Fig. 2c). Around 23% of the non-reference gene families featured

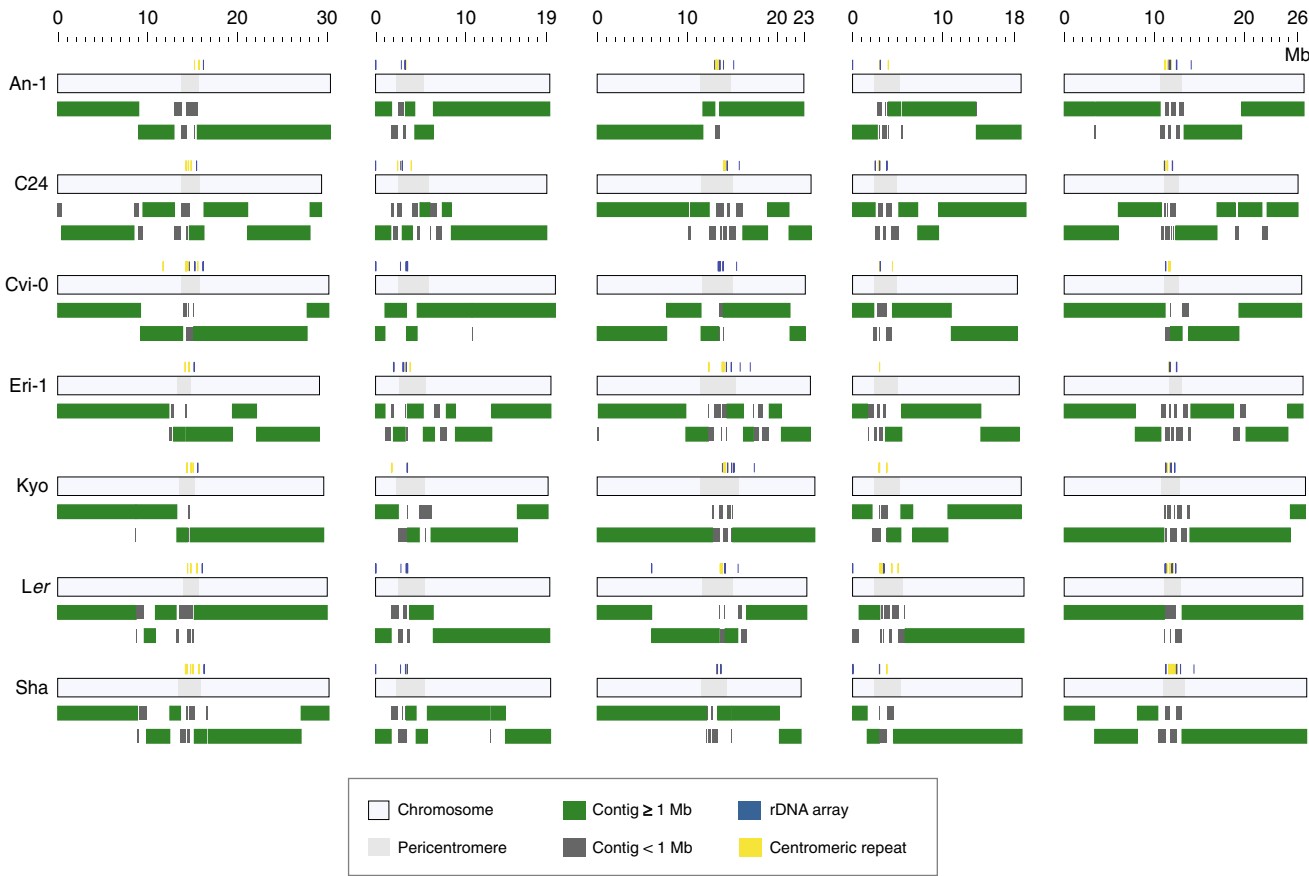

**Fig. 1 Chromosome-level genome assemblies of seven *A. thaliana* accessions.** The light gray bars outline each of the chromosomes, whereas the dark gray inlays show the extent of each of the pericentromeric regions. The contig arrangements of the chromosome assemblies is shown in green for contigs > 1 Mb and dark grey for contigs < 1 Mb. The location of centromeric tandem repeat arrays and rDNA clusters within the assemblies are marked by yellow and blue boxes above each of the chromosomes. Source Data are provided as a Source Data file.

**Table 1 Genome assembly and annotation of seven *A. thaliana* accessions.**

|  | Col-0[a] | An-1 | C24 | Cvi-0 | Eri-1 | Kyo | Ler | Sha |
|---|---|---|---|---|---|---|---|---|
| Contigs | – | 151 | 167 | 140 | 200 | 230 | 149 | 143 |
| Pseudomolecules | 5 | 5 | 5 | 5 | 5 | 5 | 5 | 5 |
| Contig N50 (Mbp) | – | 8.2 | 4.8 | 7.4 | 4.8 | 9.1 | 11.2 | 7.0 |
| Contig CL50[b] | – | 2 | 2 | 2 | 2 | 2 | 1 | 1 |
| Chr. length (Mbp) | 119.1 | 118.4 | 117.7 | 118.3 | 117.7 | 118.8 | 118.5 | 118.4 |
| Genes | 27,445 | 27,342 | 27,214 | 27,098 | 27,285 | 27,574 | 27,376 | 27,293 |

[a]Reference sequence.
[b]Chromosome number normalized L50[20].

orthologs in the closely related genome of *Arabidopsis lyrata* and, according to RNA-seq read mapping, 26–40% of them showed evidence of expression (Supplementary Table 14). The remaining 1,050 non-reference (accessions-specific) gene families were evenly distributed across the accessions (Fig. 2c), with the exception of Cvi-0, where we found nearly twice as many (214) accession-specific genes, which is in agreement with the divergent ancestry of this relict accession[8,27].

Based on all possible pairwise genome comparisons, we identified 5.1–6.5 Mb accession-specific sequence and used this to estimate a pan-genome size of ~135 Mb including ~30,000 genes and a core-genome size of ~105 Mb with ~24,000 genes (Fig. 2d)[28] illustrating that one reference genome is not sufficient to capture the entire sequence diversity within *A. thaliana*[29]. Deeper sampling including accessions from other populations

(for example by including more of the highly divergent accessions from Africa[27]) could lead to higher estimates of the pan-genome. This has been observed in short-read sequencing-based pan-genome analyses of rice and tomatoes[30,31], even though such comparisons are difficult not only due to the different samplings (even including the integration of subspecies), but also due to the high-contiguity of chromosome-level assemblies, which will reveal more of the hidden genes in complex genomic regions.

**Quantification of genome collinearity.** As only a few chromosome-level assemblies are available, hardly anything is known about genome collinearity within *A. thaliana*. In contrast, our chromosome-level assemblies allow for an analysis of the conservation of the genome collinearity between multiple

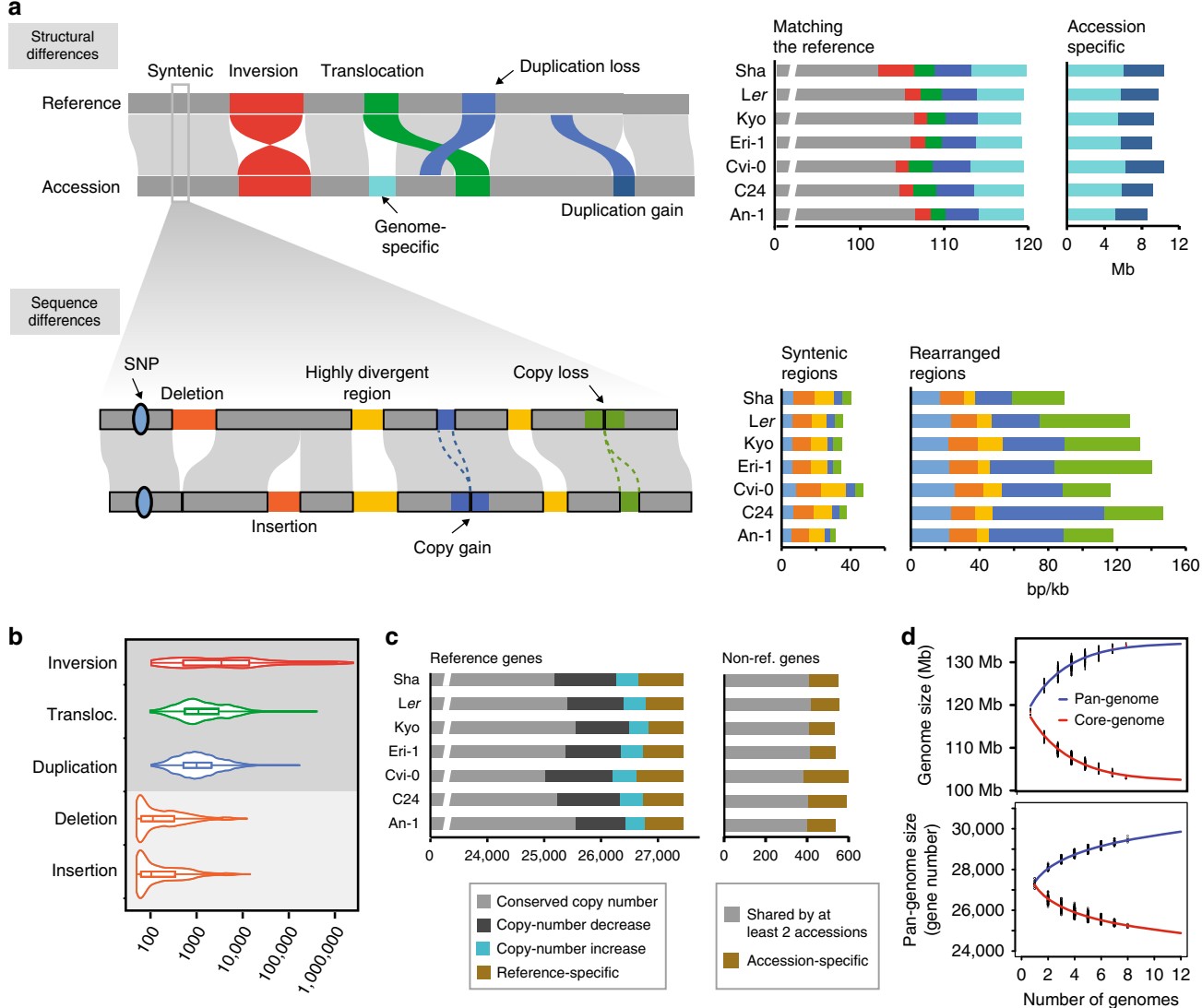

**Fig. 2 Structural and sequence differences between the genomes. a** Schematic of the structural differences (upper panel) and sequence variation (lower panel) that can be identified between chromosome-level assemblies. Note, local sequence variation can reside in syntenic as well as in rearranged regions. The barplots on the right upper side show the total span of syntenic and rearranged regions between the reference and each of other accessions (colors match the schematic on the left): The left barplots shows the sequence span in respect to the reference sequence, while the right plot shows the sequence space, which is specific to each of the accessions. The barplots on the right lower side show local sequence variation (per kb) in syntenic (left) and rearranged (right) regions between the reference and each of other accessions (again colors match the schematic on the left). **b** Size distributions of different types of structural and sequence variation. **c** Gene copy-number variations between the reference and each of the accessions. The left barplots shows the fraction of reference genes which are in gene families with conserved or variable copy numbers. The right barplots shows the number of non-reference genes found in at least two accessions, or found to be specific to an accession genome. **d** Pan-genome and core-genome estimations for sequence (upper plot) and gene space (lower plot) were based on all pairwise whole-genome and gene set comparisons across all eight accessions. Each black point corresponds to a pan- or core-genome size estimated with a particular combination of genomes. Pan-genome (blue) and core-genome (red) estimations were fitted using an exponential model. Source Data are provided as a Source Data file.

individuals. To quantify collinearity we developed a parameter called synteny diversity $\pi_{syn}$, which is similar to nucleotide diversity[32], however, instead of measuring average sequence differences it measures the degree of collinearity between the genomes of a population (see Methods). $\pi_{syn}$ values can range from 0 to 1, where 1 refers to the complete absence of collinearity between any of the genomes and 0 to regions where all genomes are collinear. $\pi_{syn}$ can be calculated in any given region; however, the annotation of collinearity still needs to be established within the context of the whole genomes to avoid false assignments of homologous but non-allelic regions.

We calculated $\pi_{syn}$ in 5-kb sliding windows across the genome using pairwise comparisons of all eight accessions (Fig. 3a). As expected, $\pi_{syn}$ was generally high in pericentromeric regions and low in chromosome arms. Overall, this revealed around 90 Mb (76% of the genome) where all genomes were collinear to each other, while for the remaining 29 Mb (24%) the collinearity between the genomes was not conserved. This, for example, included a region on chromosome 3 (ranging from Mb ~2.8–5.3), where $\pi_{syn}$ was increased to ~0.25 (i.e., one genome is not collinear to all other seven genomes) due to the 2.48 Mb inversion in the Sha genome (Fig. 3a, arrow labelled with (A)).

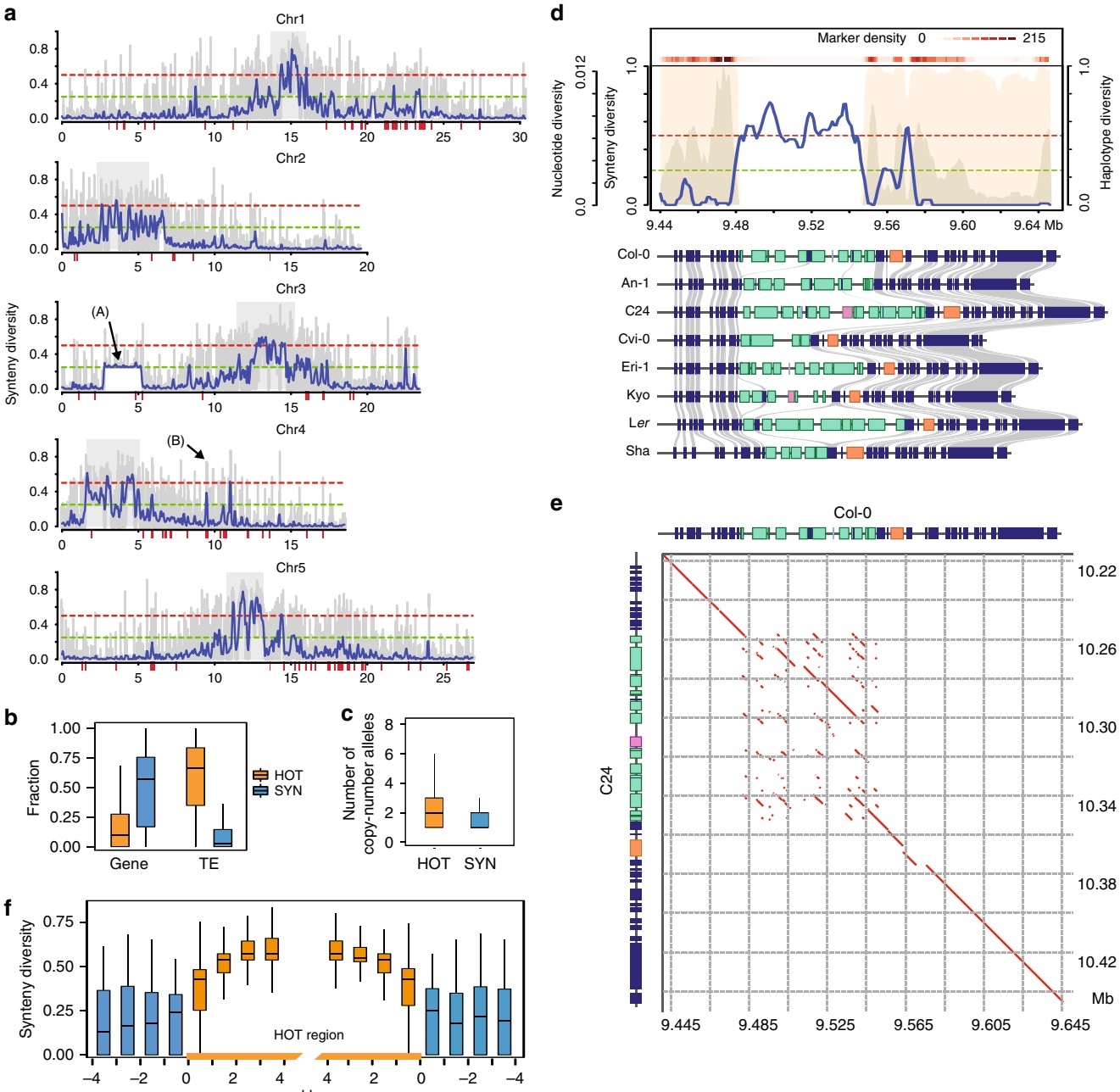

**Fig. 3 Quantitative analysis of synteny reveals hotspots of rearrangements. a** Synteny diversity along each chromosome: (100 kb sliding windows with a step-size of 50 kb in blue; 5 kb sliding windows with a step-size of 1 kb in grey). Red bars: R gene clusters. Gray rectangles: centromeres. The dashed green and red lines indicate thresholds for synteny diversity values of 0.25 and 0.50. The labelled arrow (A) indicates a 2.48 Mb inversion in the Sha genome. The labelled arrow (B) indicates the location of the example shown in **d**. **b** Gene and TE densities in 10,331 syntenic (SYN) and 576 hotspots of rearrangements (HOT) regions. **c** The number of variable copy-number alleles in 10,331 syntenic (SYN) and 576 rearrangements (HOT) regions. **d** An example of a HOT region including the *RPP4/RPP5* R gene cluster. The upper panel shows the distribution of synteny diversity (blue curve), nucleotide diversity (gray background) and haplotype diversity (pink background) in a 5 kb sliding window with a step-size of 1 kb. Both the nucleotide diversity and the haplotype diversity were calculated based on informative markers (MAF ≥ 0.05, missing rate < 0.2) from the 1001 Genomes Project[8]. The marker density is shown as the heatmap on top. The green and red dashed lines indicate the value 0.25 and 0.50 of synteny diversity, respectively. The schematic in the lower part shows the annotated protein-coding genes (colored rectangles). Blue rectangles: non-resistance genes. Other colored rectangles: resistance genes where genes with the same color belong to the same gene family. The gray links between the rectangles indicate the homologous relationships between non-resistance genes. **e** A dot plot of Col-0 and C24 sequence from the HOT region shown in **d**. Red lines: homologous regions between the two genomes. **f** The distribution of synteny diversity values in 1 kb sliding windows around and in 576 HOT regions. In box plots **b, c** and **f**, centre line: median, bounds of box: 25th and 75th percentiles, whiskers: 1.5 * IQR (IQR: the interquartile range between the 25th and the 75th percentile). Source Data are provided as a Source Data file.

**Hotspots of rearrangements.** Unexpectedly, however, some regions featured $\pi_{syn}$ values even larger than 0.5. This implied that not only two, but also multiple independent, non-collinear haplotypes segregate in these regions. In turn, this suggests that these regions are more likely to undergo or conserve complex mutations as compared to the rest of the genomes and thereby create hotspots of rearrangements (HOT regions) where multiple accessions independently evolved diverse haplotypes. Overall, we found 576 of such HOT regions with a total size of 10.2 Mb including 351 HOT regions in the gene-rich chromosome arms with a total length of 4.1 Mb (or 4% of euchromatic genome) (Supplementary Data 3).

Even though HOT regions in euchromatic regions included more transposable elements and fewer genes as compared to the collinear regions, they still contained substantial numbers of genes, many of which occurred at high and variable copy-number between the accessions (Fig. 3b, c). For example, a HOT region on chromosome 4, which overlapped with the *RPP4/RPP5* R gene cluster[33], displayed 5–15 intact or truncated copies of the *RPP5* gene across the eight genomes (Fig. 3d and Supplementary Table 15). The different gene copies were primarily introduced by an accumulation of forward tandem duplications and large indels (Fig. 3e).

This remarkable pattern of forward tandem duplications and large indels was shared by many of the HOT regions (Fig. 3c and Supplementary Fig. 4). The clear pattern of almost exclusively forward tandem duplications suggested higher mutation (duplication) rates, which are specific to these regions in each of the accessions. In contrast, the borders of the HOT regions were surprisingly well conserved across the accessions (Fig. 3f). This suggested that either different selection regimes introduced clear-cut borders between the HOT regions and their vicinity, or that HOT regions are specific targets of increased tandem duplication rates. Such a local increase of mutation rates could potentially be mediated by non-allelic homologous recombination, which could be triggered by the high number of local repeats in these regions[34]. Figure 4 shows two more examples of these complex regions.

In contrast, meiotic recombination in *Arabidopsis* was shown to be suppressed by structural diversity[35]. To test if HOTs are indeed depleted for meiotic recombination, we overlapped rearranged regions with 15,683 crossover (CO) sites previously identified within Col-0/Ler F2 progenies[35,36]. Only 64 of them partially overlapped with non-syntenic regions while all other COs were found in syntenic regions (Fig. 5a), suggesting that HOT regions are almost completely silenced for COs ($\chi^2$ test, $p < 0.001$). In consequence, this would imply that HOT regions are segregating as large non-recombining regions. To test this, we analysed the linkage disequilibrium (LD) within 1135 genomes of the 1001 Genomes Project[8] around and across the HOT regions. LD increased in the vicinity of the HOT regions, with increasing LD close to the HOT regions implying reduced recombination in the regions surrounding the HOT regions. Likewise, LD was also high within HOT regions corroborating the recombination suppression in HOT regions. However, when calculated across the border of these regions, LD was significantly lower (one-sided U test, $p < 0.001$) supporting the idea that HOT regions are not strongly linked to the surrounding haplotypes and that they hardly exchange alleles (Fig. 5b).

Reduced meiotic recombination has been linked to the accumulation of new (deleterious) mutations[37]. In agreement with this, HOT regions showed an accumulation of SNPs with low allele frequencies and potentially deleterious variation (one-sided U test, $p < 0.001$) as compared to other regions in the genome (Fig. 5c, d and Supplementary Fig. 5). Moreover, reduced recombination combined with geographic isolation can provide the basis for the development of alleles, which are incompatible with distantly related haplotypes leading to intra-species incompatibilities[38]. To test this, we searched the location of nine recently reported genetic incompatible loci[39] (*DM1-9*) and found that all except of one overlapped with HOT regions, while *DM3*, the locus which did not overlap with a HOT region, was closely flanked by two HOT regions (Figs. 3d, 4a and Supplementary Fig. 6–11). In addition, we also checked the locus of a recently published single-locus genetic incompatibility[40] and found that it was also residing in a HOT region (Supplementary Fig. 12).

The high structural diversity of the HOT regions was reminiscent of the patterns that have been described for R gene clusters[41–44]. In fact, the 808 reference genes in HOT regions were significantly enriched for genes involved in defense response, signal transduction and secondary metabolite biosynthesis (Fig. 5e) suggesting a reoccurring role of HOT regions in the adaptation to biotic stress. Further comparison with the outcrossing sister species *A. lyrata* showed that 504 (87.5%) of 576

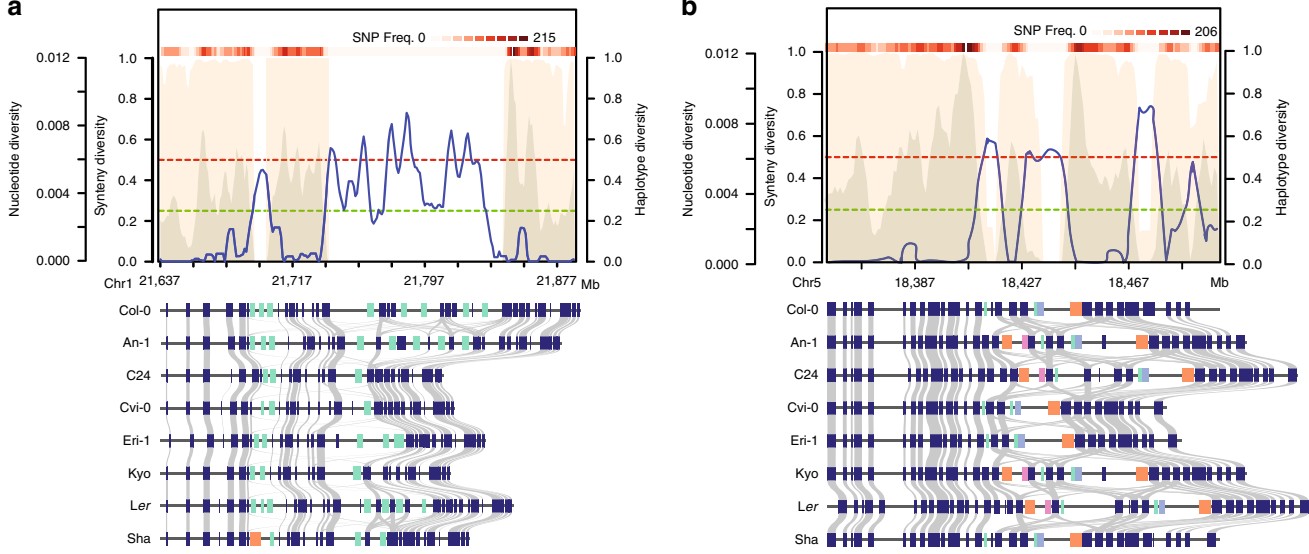

**Fig. 4 Two examples for hotspots of rearrangements.** Visualization of **a** the *DM6* locus (*RPP7*) and **b** an unnamed R gene cluster on chromosome 5. Descriptions for the plots can be found in the legend of Fig. 3d. Source Data are provided as a Source Data file.

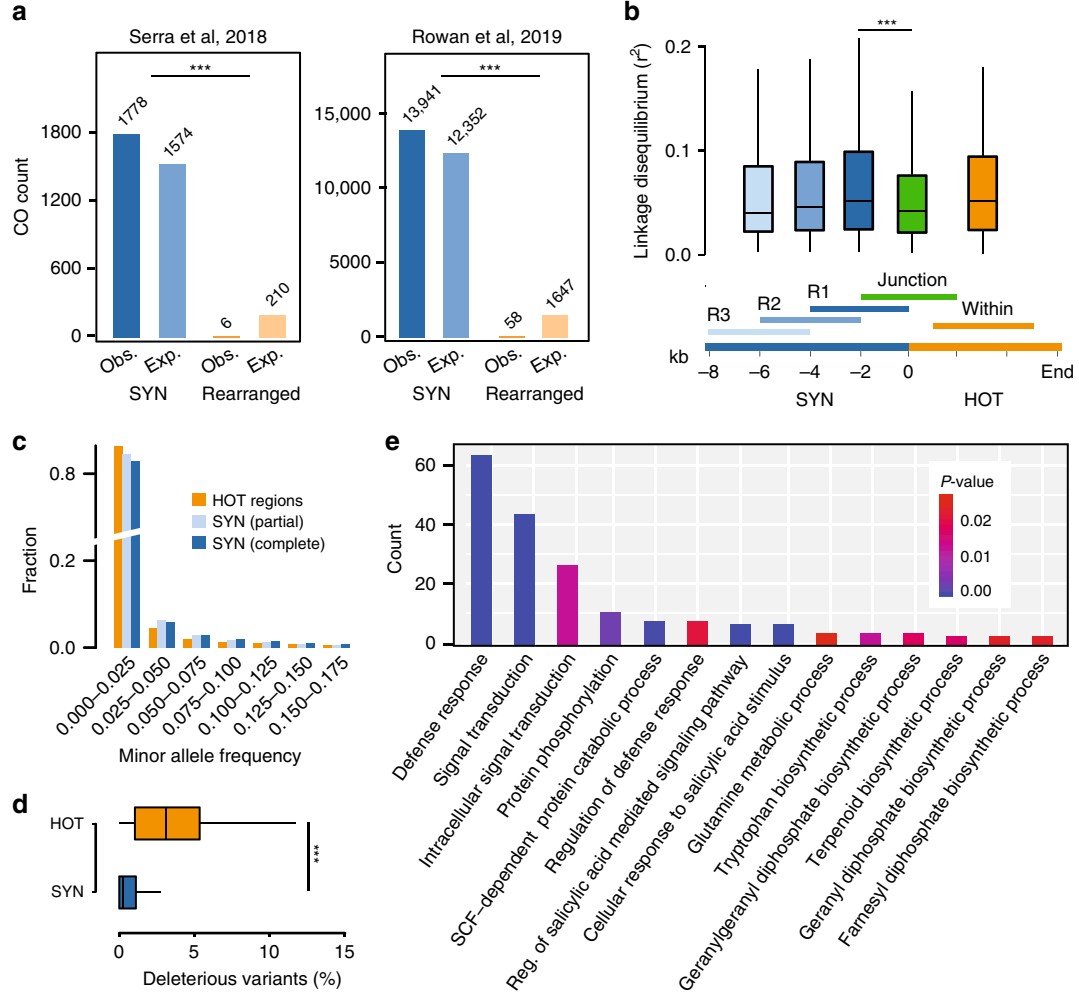

**Fig. 5 The causes and consequences of hotspots of rearrangements. a** Crossover (CO) breakpoints[35,36] identified in Col-0 x Ler hybrids were checked for their overlaps in syntenic or rearranged regions. Only unique CO intervals smaller than 5 kb were used. Obs.: observed. Exp.: expected. One-sided $\chi^2$ test was used. **b** Linkage disequilibrium (LD) calculated in 4 kb windows in and around each of the 576 HOT regions as shown in the lower part. SYN, syntenic region. LD was calculated as the correlation coefficient ($r^2$) based on informative SNP markers (MAF > 0.05, missing rate < 0.2) selected from the 1001 Genomes Project data[8]. One-sided U test was used. **c** Minor allele frequency of SNP markers in 10,331 syntenic, 10,501 partially syntenic, and 576 HOT regions. The SNP markers (MAF > 0.005, missing rate < 0.2) from 1001 Genomes Project were used. **d** Frequency of deleterious mutations in 10,331 syntenic (SYN) regions and 576 HOT regions. Deleterious mutations include SNPs and small indels that introduce premature stop codons, loss of start or stop codons, frameshifts, splicing sites mutations or deletions of exons. One-sided U test was used. **e** GO term enrichment analysis of protein-coding genes in 576 HOT regions. Fisher exact test used, $p < 0.05$. In box plots **b** and **d**, centre line: median, bounds of box: 25th and 75th percentiles, whiskers: 1.5 * IQR (IQR: the interquartile range between the 25th and the 75th percentile). $p < 0.001$: ***. Source Data are provided as a Source Data file.

HOT regions actually have no homologous sequences in A. lyrata. The flanking regions of nearly one third HOT regions remained collinear with A. lyrata, while flanking regions of the other regions are structurally rearranged, suggesting that HOT regions are likely to evolve in non-conserved regions between two species.

## Discussion

As biotic stress is an evolving environmental challenge, the Red Queen hypothesis suggests that the genomes of A. thaliana are in the constant need to diversify their offspring[45]. It has been proposed that in response to this, meiotic recombination might increase and thereby diversified offspring is generated[46]. However, exclusively shuffling existing variation might not be sufficient to respond to the evolution of pathogens. Instead, it has been proposed that the accumulation of new gene duplicates could enable a rapid genomic response of plants against

pathogens[34,47,48]. The hotspots of rearrangements have the potential to build the basis for such a response, as frequent gene duplications could build the basis for an evolutionary playground to evolve a quick response to the challenges of biotic stress and overcome fitness valleys during the evolution of more complex function. This, in turn, comes at the costs of loss of synteny and the loss of meiotic recombination between distant haplotypes. Though it still needs to be analyzed whether local populations show the same level of diversity or if their haplotypes in HOT regions are more similar and still exchange alleles, we have observed the negative consequences of reduced meiotic recombination in this small world-wide population including the accumulation of deleterious alleles and incompatible epistatic effects between distant genotypes.

Taken together, using chromosome-level genome assemblies of a small, highly diverse population of A. thaliana, we have identified regions where genome collinearity was lost through genome-specific accumulation of mutations. These quickly

evolving sequences do not spread through the population based on meiotic recombination-based exchange between haplotypes (as recombination is suppressed by structural variation) or based on haplotype-specific drift or selection (similar to an inversion allele), as the haplotypes change more rapidly than they are distributed through the population. Instead it occurs that these regions evolve through rapid mutations. We propose that these regions, which we call hotspots of rearrangements or HOT regions, facilitate evolutionary responses to rapidly changing environmental challenges and that these regions are thus undergoing different evolutionary dynamics as compared to the rest of the genome, where each region segregates with only few haplotypes. Future genome-wide screens for selection patterns should take such regions and their specific characteristics into account in particular as they might be missed with conventional marker-based selection scans.

## Methods

**Plant material and whole-genome sequencing.** We received the seeds of all seven accessions from Maarten Koornneef (MPI for Plant Breeding Research), and grew them under normal greenhouse conditions. The stock center ID of seeds are shown in the Supplementary Table 1. DNA preparation and next generation sequencing was performed by the Max Planck Genome center. DNA was extracted from multiple individuals using the NucleoSpin® Plant II Maxi Kit from Macherey-Nagel, prepared using SMRTbell Template Prep Kit 1.0-SPv3 with SMRTbell Damage Repair Kit -SPv3 and BluePippin size selection for fragments >9/10 kb, and sequenced with a PacBio Sequel system. For each accession, data from two SMRT cells were generated. Besides, Illumina paired-end libraries were prepared and sequenced on the Illumina HiSeq system.

**Genome assembly.** PacBio reads were filtered for short (<50 bp) or low quality (QV < 80) reads using SMRTLink5 package. De novo assembly of each genome was initially performed using three different assembly tools including Falcon[17], Canu[49], and MECAT[50]. The resulting assemblies were polished with Arrow from the SMRTLink5 package and then further corrected with mapping of Illumina short reads using BWA[51] to remove small-scale assembly errors which were identified with SAMTools[52]. For each genome, the final assembly was based on the Falcon assembly as these assemblies always showed highest assembly contiguity. A few contigs were further connected or extended based on whole-genome alignments between Falcon and Canu or MECAT assemblies. Contigs were labelled as organellar contigs if they showed alignment identity and coverage both larger than 95% when aligned against the mitochondrial or chloroplast reference sequences. A few of contigs aligned to multiple chromosomes and were split if no Illumina short-read alignments supported the conflicting regions. Assembly contigs larger than 20 kb were combined to pseudo-chromosomes according to their alignment positions when aligned against the reference sequence using MUMmer4[53]. Contigs with consecutive alignments were concatenated with a stretch of 500 Ns. To note, the assembly of the Ler accession was already described in a recent study[13].

**Assembly evaluation.** We evaluated the assembly completeness by aligning the reference genes against each of the seven genomes using Blastn[54]. Reference genes which were not aligned or only partially aligned might reveal genes which were missed during the assembly. To examine whether they were really missed, we mapped Illumina short reads from each genome against the reference genome using the BWA[51] and checked the mapping coverage of these genes. The genes, which were missing in the assembly, should show full alignment coverage (Supplementary Table 7).

Centromeric and telomeric tandem repeats were annotated by searching for the 178 bp tandem repeat unit[55] and the 7 bp tandem repeat unit of TTTAGGG[56]. rDNA clusters were annotated with Infernal version 1.1[57].

The assembly contiguity of Cvi-0 and Ler were further tested using three previously published genetic maps[24,58,59] (Supplementary Table 4). For this we aligned the marker sequences against the chromosome-level assemblies and checked the order of the markers in the assembly versus their order in the genetic map. The ordering of contigs to chromosomes was perfectly supported by all three maps. Overall, only six (out of 1156) markers showed conflicts between the genetic and physical map. In all six cases we found evidence that the conflict was likely caused by structural differences between the parental genomes.

We also searched for potentially collapsed regions in each assembly. For this, we checked the normalized mapping coverage in non-overlapping 100 bp windows based on Illumina short-read mapping (using BWA). Collapsed regions are expected to have significantly higher coverage than the correctly assembled regions. Here, windows with two-fold increase of mapping coverage were defined as collapsed regions. Continuous collapsed regions were merged.

**Gene annotation.** Protein-coding genes were annotated based on ab initio gene predictions, protein sequence alignments and RNA-seq data. Three ab initio gene prediction tools were used including Augustus[60], GlimmerHMM[61] and SNAP[62]. The reference protein sequences from the Araport 11[21] annotation were aligned to each genome assembly using exonerate[63] with the parameter setting "–percent 70–minintron 10–maxintron 60000". For five accessions (An-1, C24, Cvi-0, Ler, and Sha) we downloaded a total of 155 RNA-seq datasets from the NCBI SRA database (Supplementary Data 2). RNA-seq reads were mapped to the corresponding genome using HISAT2[64] and then assembled into transcripts using StringTie[65] (both with default parameters). All different evidences were integrated into consensus gene models using Evidence Modeler[66].

The resulting gene models were further evaluated and updated using the Araport 11[21] annotation. Firstly, for each of the seven genomes, the predicted gene and protein sequences were aligned to the reference sequence, while all reference gene and protein sequences were aligned to each of the other seven genomes using Blast[54]. Then, potentially mis-annotated genes including mis-merged (two or more genes are annotated as a single gene), mis-split (one gene is annotated as two or more genes) and unannotated genes were identified based on the alignments using in-house python scripts. Mis-annotated or unannotated genes were corrected or added by incorporating the open reading frames generated by ab initio predictions or protein sequence alignment using Scipio[67].

Noncoding genes were annotated by searching the Rfam database[68] using Infernal version 1.1[57]. Transposon elements were annotated with RepeatMasker (http://www.repeatmasker.org). Disease resistance genes were annotated using RGAugury[69]. NB-LRR R gene clusters were defined based on the annotation from a previous study[70].

**Pan-genome analysis.** Pan-genome analyses were performed at both sequence and gene level. To construct a pan-genome of sequences, we generated pairwise whole-genome sequence alignments of all possible pairs of the eight genomes using the nucmer in the software package MUMmer4[53]. A pan-genome was initiated by choosing one of the genomes, followed by iteratively adding the non-aligned sequence of one of the remaining genomes. Here, non-aligned sequences were required to be longer than 100 bp without alignment of more than 90%. The core genome was defined as the sequence space shared by all sampled genomes. Like the pan-genome, the core-genome analysis was initiated with one genome. Then all other genomes were iteratively added, while excluding all those regions, which were not aligned against each of the other genomes. The pan- and core-genome of genes was built in a similar way. The pan-genome of genes was constructed by selecting the whole protein-coding gene set of one of the accessions followed by iteratively adding the genes of one of the remaining accessions. Likewise, the core-genome of genes was defined as the genes shared in all sampled genomes.

For each pan or core-genomes analysis, all possible combinations of integrating the eight genomes (or a subset of them) were evaluated (Eq. 1). The exponential regression model (Eq. 2) was then used to model the pan-genome/core-genomes by fitting medians using the least square method implemented in the nls function of R.

$$\left( \sum_{n=1}^{8} \left( \frac{8!}{n!(8-n)} \right) \right) \qquad (1)$$

$$y = Ae^{Bx} + C \qquad (2)$$

**Analysis of structural rearrangements and gene CNV.** All assemblies were aligned to the reference sequence using nucmer from the MUMmer4[53] toolbox with parameter setting "-max -l 40 -g 90 -b 100 -c 200". The resulting alignments were further filtered for alignment length (>100) and identity (>90). Structural rearrangements and local variations were identified using SyRI[13]. The functional effects of sequence variation were annotated with snpEff[71]. The gene CNV were identified according to the gene family clustering using the tool OrthoFinder[26] based on all protein sequences from the eight accessions.

**Definition of synteny diversity.** Synteny diversity was defined as the average fraction of non-syntenic sites found within all pairwise genome comparisons within a given population. Here we denote synteny diversity as (Eq. 3)

$$\pi_{syn} = \sum_{ij} x_i x_j \pi_{ij}, \qquad (3)$$

where $x_i$ and $x_j$ refer to the frequencies of sequence $i$ and $j$ and $\pi_{ij}$ to the average probability of a position to be non-syntenic between sequence $i$ and $j$. Note, $\pi_{syn}$ can be calculated in a given region or for the entire genome. However even when calculated for small regions the annotation of synteny still needs to be established within the context of the whole genomes to avoid false assignments of homologous but non-allelic sequence. Here we used the annotation of SyRI to define syntenic regions. $\pi_{syn}$ values can range from 0 to 1, with higher values referring to a higher average degree of non-syntenic regions between the genomes.

**Analysis of hotspots of rearrangements**. For the analyses, we calculated $\pi_{syn}$ in 5-kb sliding windows with 1 kb step-size across the entire genome. HOT regions were defined as regions with $\pi_{syn}$ larger than 0.5. Neighboring regions were merged into one HOT region if their distance was shorter than 2 kb.

The nucleotide and haplotype diversity were calculated with the R package PopGenome[72] using SNP markers (with MAF > 0.05) from 1001 Genomes Project[8]. LD were calculated as correlation coefficients $r^2$ using SNP markers with MAF > 0.05. GO enrichment analysis was performed using the webtool DAVID[73].

We performed a synteny comparison between *A. thaliana* HOT regions and *A. lyrata*[74]. Although the two species have rearranged karyotypes, they share collinear regions, so-called Ancestral Crucifer Karyotype blocks (ACK blocks)[75]. The genome sequences were split into ACK blocks, and aligned with the tool nucmer. The syntenic regions were defined by the tool SyRI. We checked the alignment of each HOT region and its 5 kb flanking regions to see whether the regions were in syntenic or rearranged regions as compared to *A. lyrata*.

**Reporting summary**. Further information on research design is available in the Nature Research Reporting Summary linked to this article.

## Data availability

Data supporting the findings of this work are available within the paper and its Supplementary Information files. The datasets generated and analyzed during the current study are available from the corresponding author upon request. Raw sequencing data, assemblies and annotations can be accessed in the European Nucleotide Archive under the project accession number PRJEB31147. Assemblies, annotation, variation and orthologs can be found on the 1001 *Arabidopsis thaliana* Genomes webpage [https://1001genomes.org/data/MPIPZ/MPIPZJiao2020/releases/current/]. Previously reported RNA-seq data from the five accessions (An-1, C24, Cvi-0, L*er*, and Sha) are downloaded from the NCBI SRA database (the NCBI and ENA accession codes are included in Supplementary Data 2). The SNP markers resulted from 1001 Genomes Project can be downloaded from the webpage https://1001genomes.org/data/GMI-MPI/releases/v3.1/. The source data underlying Figs. 1–5 and Supplementary Figs. 1–12 are provided as a Source Data file.

## Code availability

Custom code used in this study can be freely accessed at https://github.com/schneebergerlab/AMPRIL-genomes.

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

## Acknowledgements

The authors would like to thank Beth A. Rowan (UC Davis) for providing the CO breakpoint list prior to publication, Bruno Hüttel (Max Planck Genome center) for support in genome sequencing, Sigi Effgen and Maarten Koornneef (Max Planck Institute for Plant Breeding Research) for providing seeds, Onur Dogan (Max Planck Institute for Plant Breeding Research) for help in the greenhouse, Angela M. Hancock (Max Planck Institute for Plant Breeding Research) for helpful discussions, and Raphael Mercier and Padraic J. Flood (Max Planck Institute for Plant Breeding Research) for helpful comments on the manuscript and the interpretation of HOT regions. K.S. gratefully acknowledges support from European Research Council (ERC) Grant "INTERACT" (802629).

## Author contributions

W.-B.J. and K.S. designed the study. W.-B.J. performed all analysis. K.S. supervised the study. W.-B.J. and K.S. wrote the manuscript. All authors read and approved the final manuscript.

## Competing interests

The authors declare no competing interests.
