## [Peer Review File · Nature Communications]

Reviewers' comments:

Reviewer #1 (Remarks to the Author):

I read this report from Jiao and Schneeberger with great interest. The discovery and description of these 'HOT' regions is important for those working in systems in diverse kingdoms, although it is situated in *Arabidopsis thaliana*, an overwhelmingly selfing plant. The suggestion of a trade-off between the potential adaptive benefits of these HOT regions and the costs of synteny loss is absolutely fascinating and merits development. In the scope of this MS, however, I believe exactly enough is presented, and in a digestible format, as is. Overall, I found this work to be very clear, well-executed, and convincing.

I am very curious about the exciting phrases in the abstract (lns 25-28) about the 'altered evolutionary dynamics' of the HOT regions enabling 'a quick response to the ever-evolving challenges' and I would like to see some more evolutionary analysis of selection in these regions if possible. This explicitly evolutionary perspective features prominently in the abstract and conclusion, but the results and discussion do not maintain this tone so much. I'm not necessarily suggesting more analysis, but it might be useful to flag the topic when discussing the impacts of varying synteny diversity, a clever metric novel to this MS (and the fact that these are truly 'big hit' mutations). But I understand that there's already a lot in this MS and that much discussion of this may be beyond the scope of this paper.

One addition that I think would boost the impact of the study would be to include some brief discussion of the broader work on such global assessments of structural variation or pangenome work in other kingdoms, or where there's little relevant work, also in plants. For example, the 3,000 rice pangenome paper [Wang et al Nature 2018] could be related to this work in certain places (e.g. in discussing proportions of new genes discovered with deeper sampling, though I acknowledge of course that the Wang MS used an inferior Illumina approach; in any case this can help show the power of Jiao's approach and potentially, broader patterns across systems, that in this case are bottlenecked.).

There were some small typos and errors, which I have noted below, along with minor wording suggestions the authors might consider:

Ln. 30: In some sense I do like the equation of 'exchange of alleles' to '(chromosome arms)', but as I read it there's an implication that CO are the only/major way this occurs, which would downplay gene conversion, a much more common phenomenon that changes allele frequencies in one step (but with albeit a smaller footprint per event). I wonder if removing '(chromosome arms)' is better here?

Ln. 31: 'the time': do the authors mean 'the same time'?

Ln. 33: 'importance to preserved: do the authors mean 'importance of preserving'?

Ln. 47: 'there are only few: do the authors mean 'there are very few'? Or can they be more accurate in describing the state of work in plants and elsewhere for this generalist journal?

Ln. 564: 'extend' to 'extent'

Ln. 113-4: This is a small point but the sentence 'As most of the *A. thaliana* genomes have been analyzed with short read sequencing hardly anything is known about their collinearity' could be better phrased: it's not because most of the genomes have been analysed with short reads: if a minority of the 1135 *thaliana* (say, 200) were chromosome build and 935 were not, quite a lot indeed would be known about their collinearity. I'd suggest simply striking the first part of the phrase and just stating not much is known about collinearity as there have been not enough

contiguous assemblies and dedicated study. Also 'in contract': I think you mean 'in contrast'.

Ln. 141: I would suggest modifying 'significant numbers of genes' 'substantial numbers of genes'.

Ln. 595 'show' to 'shown'

Ln. 159 'HRs' obviously means HOT regions, but it would be helpful to define it here at first mention.

I feel the authors are too greatly understating the importance of their work in the closing sentence. I would change the 'might' to 'should' and go farther by suggesting that these HOT regions be focussed upon in selection scans.

All in all, this is a truly impressive manuscript and one which I am certain will be of importance broadly.

Reviewer #2 (Remarks to the Author):

To Arabidopsis researchers, these new and high-quality genomes represent a valuable community resource, in addition to their novel description of inter-accession rearrangements.

Can the authors comment on how the degree of fragmentation of these genomes compares with the other (long-awaited) genomes generated by other members of the 1001 Genomes Consortium, or are those data not yet available?

Just a few comments:

It is generally accepted that the Col-0 reference genome contains collapsed regions where tandem repeats (genetic or otherwise) were not resolved by the BAC-based Sanger method.

How many such collapsed regions did they find in the Col-0 genome. How do these contribute to their assessments of non-collinearity etc? Are they confident that their own assembly methods do not collapse repeats.

More generally did the authors find evidence for likely errors in the current Col-0 assembly that were perhaps supported by all of their sequenced genomes?

The seed used in this study should be described with the unique identifiers used by the EU and US stock centers.

264, 281: correct predication to prediction

273-280: It would be of interest to know how many potential errors in the Araport11 Col-0 annotation were observed during this multi-accession comparison phase.

290: Which accession was used to initiate the pan-genome – genomes or genes?

Reviewer #3 (Remarks to the Author):

In this manuscript, the authors aim to provide a pan-genome view of the Arabidopsis thaliana by comparing multiple genomes of the species with reference-quality genome assemblies. The dark side of the plant genomes, which I refer to highly repetitive regions, has often been obscured in numerous genome sequencing projects due to technical limitation innate in currently preferred short-read based sequencing platforms. The authors combined the usage of PacBio and Illumina whole-genome sequencing to achieve impressive depth to assemble seven genomes of the species. With this dataset, they for the first time could provide us with the "estimated size" of the core- vs and pan-genome respectively. This is a milestone research relevant not only for the A.

thaliana community but also for population genomics attempting to construct high-quality multiple references regardless of species of choice. I evaluate that this study is timely and deserves immediate attention from the field, the notion which my lab members and myself already appreciated from their BioRxiv preprint. The study emphasised the need for more than a reference-quality single genome for a species, something that was hinted at in a recently published paper by Van de Weyer et al. (2019) that revealed a large excess of novel NLR-type immune receptor encoding genes not present in the reference Col-0 genome.

The analytical tools that the authors employed to identify regions deviating from collinearity set the stage for the follow-up studies. Thus the pan-genome information derived from the seven genomes will certainly be invaluable in guiding the field to grasp variability across a large number of accessions especially in the regions with low synteny support. *nsyn*, a metric developed in this study, is also valuable in that it allows conservation to be quantified across highly variable regions that may not be amenable to conventional measurements (such nucleotide diversity, see Fig. 3d, 4 of the paper). I enjoyed reading this manuscript, as the analysis revealed the propensity of immune gene rearrangements in a genome-wide, quantitative manner, which had only been anecdotally reported in numerous plant species. Their graphic presentation of DM loci with multi-gene NLR clusters is quite citable as textbook information. Given that information of such important loci in numerous crop species, often conferring disease resistance, is missing, the provided pan-genome view on immune clusters offers a way to design molecular markers linked to rearrangements.

With the impressive dataset, the authors could potentially explore further to provide answers to the following questions: 1) Do the HOT regions in *A. thaliana* overlap with the synteny breaking regions between *A. thaliana* and other sister species? It would be interesting to see how the extent and types of conservation (or loss) of synteny within *A. thaliana* compares with cross-species comparisons. The pattern of variation to be found between *A. thaliana* and inbred sisters vs between *A. thaliana* and outcrossing relatives might tell us potential contribution of prevalent rearrangements near immune genes in relation to breeding strategies; 2) How reliable is the LD analysis around and across HOT regions from 1,135 genomes given that the SNP data comes from short-reads? I wonder if the LD comparisons that authors used are valid considering potential SNP acquisition bias. For example, SNP obtained within HOT regions would be less representative than the rest of the region under queries.

Minor comments are as follow.

Line 89

Salome PA et al. 2012 (Pubmed ID 22072068) should be added. This work clearly stated the inversion on Chromosome 3 in Sha.

Line 159

HR shall be spelled out as HOT region since this is the only place where the authors used a shorten form. HR is often used to refer the hypersensitive response in plant immunity research.

Reference 48 needs to be updated with full citation.

The following lines need to be checked for typos: Line 31 (at the same time), 73 (double usage of ,which), 82 (abundantly), 114 (In contrast), 249 (full).

Regarding Figure 4 and other supplementary Figures visualizing DM loci, authors may check if the width of NLR genic regions are properly scaled. Compared to Col-0 ones, the ones in other accessions generally look slimmer. It would be also good to state if each colored region was curated for protein coding such as early stop codon (and thus become slimmer) or merely coloring shared genic regions regardless of protein sequences.

In the methods, it should be stated whether or not the authors extracted DNA from single individuals or pooled samples. Since there are multiple versions of accessions circulating among research groups, the authors may consider stating their corresponding CS # and/or depositing their materials to stock center if single seed descendants are available.

Reviewers' comments:

Reviewer #1 (Remarks to the Author):

*I read this report from Jiao and Schneeberger with great interest. The discovery and description of these 'HOT' regions is important for those working in systems in diverse kingdoms, although it is situated in *Arabidopsis thaliana*, an overwhelmingly selfing plant. The suggestion of a trade-off between the potential adaptive benefits of these HOT regions and the costs of synteny loss is absolutely fascinating and merits development. In the scope of this MS, however, I believe exactly enough is presented, and in a digestible format, as is. Overall, I found this work to be very clear, well-executed, and convincing.*

Thank you very much for your very supportive evaluation.

I am very curious about the exciting phrases in the abstract (lns 25-28) about the 'altered evolutionary dynamics' of the HOT regions enabling 'a quick response to the ever-evolving challenges' and I would like to see some more evolutionary analysis of selection in these regions if possible. This explicitly evolutionary perspective features prominently in the abstract and conclusion, but the results and discussion do not maintain this tone so much. I'm not necessarily suggesting more analysis, but it might be useful to flag the topic when discussing the impacts of varying synteny diversity, a clever metric novel to this MS (and the fact that these are truly 'big hit' mutations). But I understand that there's already a lot in this MS and that much discussion of this may be beyond the scope of this paper.

Thank you for highlighting the importance of the impact of selection on the HOT regions. The application of "conventional" maker-based selection scans might not be effective as they typically rely on the presence of syntenic or at least homologous sequence, which both is complicated in the HOT regions, where the huge redundancy in the clusters does not allow for unique assignments/alignments between individual regions. Using the regions next to HOT regions instead as a proxy to estimate selection on HOT regions is challenged by reduced linkage between these regions.

To address their evolutionary dynamics, we have highlighted/analyzed the presence of enormous sequence and gene copy variation in these regions, which (using their functional annotation as a basis) does not appear random (but selected). These quickly evolving sequences however do not spread in the common sense through the population either based on meiotic recombination-based exchange between haplotypes or based on haplotype-specific selection (like the selection of an inversion allele). Instead these regions appear specific for each genome and all of this together is what we refer to when we suggest "altered evolutionary dynamics".

One addition that I think would boost the impact of the study would be to include some brief discussion of the broader work on such global assessments of structural variation or pangenome work in other kingdoms, or where there's little relevant work, also in plants. For example, the 3,000 rice pangenome paper [Wang et al Nature 2018] could be related to this work in certain places (e.g. in discussing proportions of new genes discovered with deeper sampling, though I acknowledge of course that the Wang MS used an inferior Illumina approach; in any case this can help show the power of Jiao's approach and potentially, broader patterns across systems, that in this case are bottlenecked.).

Thank you for this suggestion to include a broader view on pan-genomes including the analyses in other systems. We have added a paragraph after the describing the results of our pangenome analysis: "Deeper sampling including accessions from other populations (for example more of the highly divergent accessions from Africa (Durvasula et al., PNAS, 2017)) could lead to higher estimates of the pan-genome. This has been observed in short read sequencing-based pangenome analyses of rice and tomatoes (Wang et al., Nature, 2018; Gao et al., Nat. Genet, 2019), even though such comparisons are difficult not only due to the different samplings (including the integration of multiple subspecies in each of these studies), but also due to the high-contiguity of

chromosome-level assemblies, which will reveal more of the hidden genes in complex genomic regions.”

Inferring broader patterns across systems (based on the available studies) is complicated not only as different sequencing technologies were used, but also by the different samplings including the selection of divergent sub-species (which might introduce different kind of variation) and it might take some more further efforts to unify the data until broad patterns between species can be inferred.

There were some small typos and errors, which I have noted below, along with minor wording suggestions the authors might consider:

Ln. 30: In some sense I do like the equation of ‘exchange of alleles’ to ‘(chromosome arms)’, but as I read it there’s an implication that CO are the only/major way this occurs, which would downplay gene conversion, a much more common phenomenon that changes allele frequencies in one step (but with albeit a smaller footprint per event). I wonder if removing ‘(chromosome arms)’ is better here?

Agreed, we removed “(chromosome arms)”.

Ln. 31: ‘the time’: do the authors mean ‘the same time’?

Corrected.

Ln. 33: ‘importance to preserved: do the authors mean ‘importance of preserving’?

Corrected.

Ln. 47: ‘there are only few: do the authors mean ‘there are very few’? Or can they be more accurate in describing the state of work in plants and elsewhere for this generalist journal?

We have added a summary of all published *de novo* assemblies of different *A. thaliana* accessions that we are aware of. As this paragraph already zoomed in on Arabidopsis we prefer to limit this to Arabidopsis only.

Ln. 564: ‘extend’ to ‘extent’

Corrected.

*Ln. 113-4: This is a small point but the sentence ‘As most of the *A. thaliana* genomes have been analyzed with short read sequencing hardly anything is known about their collinearity’ could be better phrased: it’s not because most of the genomes have been analysed with short reads: if a minority of the 1135 thaliana (say, 200) were chromosome build and 935 were not, quite a lot indeed would be known about their collinearity. I’d suggest simply striking the first part of the phrase and just stating not much is known about collinearity as there have been not enough contiguous assemblies and dedicated study.*

Good point, we agree. We changed the sentence to “As only a few chromosome-level assemblies are available, hardly anything is known about genome collinearity within *A. thaliana*.”.

Also ‘in contract’: I think you mean ‘in contrast’.

Corrected.

Ln. 141: I would suggest modifying ‘significant numbers of genes’ ‘substantial numbers of genes’.

Modified as suggested.

Ln. 595 'show' to 'shown'

Corrected.

Ln. 159 'HRs' obviously means HOT regions, but it would be helpful to define it here at first mention.

Corrected.

I feel the authors are too greatly understating the importance of their work in the closing sentence. I would change the 'might' to 'should' and go farther by suggesting that these HOT regions be focussed upon in selection scans.

Thank you for this supportive comment, we agree that the combined nature of being hard-to-analyze and being a putative target of selection makes these regions important for any kind of selection scans. In addition, just using linkage with markers in syntenic regions has the danger to miss HOTS regions. We therefore changed the last sentence to "Future genome-wide screens for selection patterns should take such regions into account in particular as they might be missed with conventional marker-based selection scans."

All in all, this is a truly impressive manuscript and one which I am certain will be of importance broadly.

Once more, thank you very much for this very supportive review.

Reviewer #2 (Remarks to the Author):

To Arabidopsis researchers, these new and high-quality genomes represent a valuable community resource, in addition to their novel description of inter-accession rearrangements.

Can the authors comment on how the degree of fragmentation of these genomes compares with the other (long-awaited) genomes generated by other members of the 1001 Genomes Consortium, or are those data not yet available?

This certainly would be an interesting comparison, however, so far there aren't any long-read-based genome assemblies of the 1001 Genomes Consortium published. Available long-read genome assemblies include a re-assembly of the reference accession Col-0 as well as assemblies of four different accessions including Cvi-0, KBS-Mac-74, Ler and Nd-1 which have been generated in different studies and have not been compared against each other. Their assembly contig N50 values were 7.4 Mb (Col-0), 6.1Mb (Cvi-0), 12.3 Mb (KBS-Mac-74), 11.2 Mb (Ler) and 13.4 Mb (Nd-1), which is similar to the contig N50 values (4.8-11.2 Mb) of this study, where we additionally mapped the contigs the chromosomes.

Just a few comments:

It is generally accepted that the Col-0 reference genome contains collapsed regions where tandem repeats (genetic or otherwise) were not resolved by the BAC-based Sanger method. How many such collapsed regions did they find in the Col-0 genome. How do these contribute to their assessments of non-collinearity etc? Are they confident that their own assembly methods do not collapse repeats. How many in Col-0 genome?

We agree that even the high-quality sequence of the Col-0 reference accession has collapsed regions, which can mostly be observed in the pericentromeric regions. As we have not re-assembled Col-0, we have not specifically searched for any issues with the reference sequence. We have now analyzed the degree of collapsed regions in the individual assemblies by realigning the short reads (allowing for unique alignments only). This reveals collapsed regions as spike/peaks in this coverage along the chromosomes. We can see that these peaks accumulate in the peri-centromere, pointing out that the assemblies have similar issues as the reference and most of the other assemblies as well. This analysis is now mentioned in the manuscript. To note, however, these regions have no effect on our synteny analysis which is focused on the chromosome arm regions.

To identify collapsed regions, we checked the fold change of normalized mapping coverage in non-overlapping continuous 100bp window based on Illumina short read mapping against the respective assemblies. Collapsed regions are expected to have significantly higher coverage than the correctly assembled regions. Here, we defined each window with a two-fold increase of mapping coverage as a collapsed region, where neighboring collapsed regions were merged. Within the reference sequence, we identified 326 collapsed regions (509 kb), where only 6% of them were in the chromosome arms. The total sizes of collapsed regions in the seven assemblies were smaller than in the reference genome. The number and total size of potentially collapsed regions are shown as below.

Accession	Number of collapsed regions	Cumulative size (bp)
An-1	288	187,700
C24	178	106,500
Col-0	326	508,800
Cvi-0	209	211,300
Eri-1	144	99,200
Kyo	187	115,200
Ler	294	260,500
Sha	485	507,800

More generally did the authors find evidence for likely errors in the current Col-0 assembly that were perhaps supported by all of their sequenced genomes?

As we mentioned above, we found 326 (509 kb) potentially collapsed regions. Besides, 2,138 small differences could be identified in the Illumina short read alignments of our Col-0 sample to the Col-0 reference. These small variations might be due to assembly errors or due to mutations between the line used for reference assembly and our Col-0 (as it was described for differences between different “Ler” samples (Zapata et al, 2016, PNAS)).

Although the comparisons between Col-0 and our seven genomes revealed some sequence difference specific to Col-0, we cannot conclude that those result from assembly errors as they might be specific variation to Col-0. However, we did find that the flanking regions of 41 of the 70 gaps (i.e. N stretches with more than 50 Ns) in Col-0 reference could be fully aligned by single contigs in our assemblies suggesting that the Col-0 assembly can soon be improved using a simple long-read assembly similar to ours.

The seed used in this study should be described with the unique identifiers used by the EU and US stock centers.

Agree, we have added this information to supplementary table S1.

264, 281: correct predication to prediction

Thanks, corrected.

273-280: It would be of interest to know how many potential errors in the Araport11 Col-0 annotation were observed during this multi-accession comparison phase.

Here we used the Araport11 Col-0 reference annotation to improve and correct our protein-coding gene models resulting from primary annotation pipeline. With this we tried to decrease false identification of gene copy number variations due to the mis-annotation and/or unannotated genes. We have not used the new annotations to correct “back” the gene models during this phase. However, we did find 35 protein-coding gene models in Araport11, which do not have a coding region with a length of a multiplier of three. Besides, 24 super short (coding region < 60 bp) protein-coding genes are present in the Araport11 annotation. It is hard to decide whether these are truly wrong, even though they are even shorter than the short open read frames (sORFs) genes that have been described earlier (Hanada et al., PNAS, 2013). However, surprisingly, we did find 14 genes which encode for protein sequences that were even shorter than ten amino acids. For example, the gene AT1G64633 codes for a peptide consisting of only one amino acid.

290: Which accession was used to initiate the pan-genome – genomes or genes?

We have calculated the pan-genome (and the remaining core space) for each possible sequence of the eight accessions. Hence, we used each of eight genomes to initiate the pan-genome (see Methods).

Reviewer #3 (Remarks to the Author):

In this manuscript, the authors aim to provide a pan-genome view of the Arabidopsis thaliana by comparing multiple genomes of the species with reference-quality genome assemblies. The dark side of the plant genomes, which I refer to highly repetitive regions, has often been obscured in numerous genome sequencing projects due to technical limitation innate in currently preferred short-read based sequencing platforms. The authors combined the usage of PacBio and Illumina whole-genome sequencing to achieve impressive depth to assemble seven genomes of the species. With this dataset, they for the first time could provide us with the “estimated size” of the core- vs and pan-genome respectively. This is a milestone research relevant not only for the A. thaliana community but also for population genomics attempting to construct high-quality multiple references regardless of species of choice. I evaluate that this study is timely and deserves immediate attention from the field, the notion which my lab members and myself already appreciated from their BioRxiv preprint. The study emphasised the need for more than a reference-quality single genome for a species, something that was hinted at in a recently published paper by Van de Weyer et al. (2019) that revealed a large excess of novel NLR-type immune receptor encoding genes not present in the reference Col-0 genome.

Thank you very much for your kind evaluation, and for pointing out this relevant reference, which we had cited initially, which however got lost during the several rounds of reformatting of the manuscript. We have added the reference back to the manuscript.

The analytical tools that the authors employed to identify regions deviating from collinearity set the stage for the follow-up studies. Thus the pan-genome information derived from the seven genomes will certainly be invaluable in guiding the field to grasp variability across a large number of accessions especially in the regions with low synteny support. π syn, a metric developed in this study, is also valuable in that it allows conservation to be quantified across highly variable regions that may not be amenable to conventional measurements (such nucleotide diversity, see Fig. 3d, 4 of the paper). I enjoyed reading this manuscript, as the analysis revealed the propensity of immune gene rearrangements in a genome-wide, quantitative manner, which had only been anecdotally reported in numerous plant species. Their graphic presentation of DM loci with multi-gene NLR clusters is quite citable as textbook information. Given that information of such important loci in numerous crop species, often conferring disease resistance, is missing, the provided pan-genome view on immune clusters offers a way to design molecular markers linked to rearrangements.

With the impressive dataset, the authors could potentially explore further to provide answers to the following questions: 1) Do the HOT regions in A. thaliana overlap with the synteny breaking regions between A. thaliana and other sister species? It would be interesting to see how the extent and types of conservation (or loss) of synteny within A. thaliana compares with cross-species comparisons. The pattern of variation to be found between A. thaliana and inbred sisters vs between A. thaliana and outcrossing relatives might tell us potential contribution of prevalent rearrangements near immune genes in relation to breeding strategies;

Thank you for bringing up this interesting analysis. In response to this point, we performed a whole genome alignment between the reference sequencing and the reference assembly of the outcrossing sister species *Arabidopsis lyrata*, and defined syntenic region using SyRI.

This revealed that 504 (87.5%) of 576 HOT regions actually have no homologous sequences between the species (defined with blastn identity > 80%). The flanking regions of nearly one third HOT regions can be found in collinear regions in *A. lyrata*, while flanking regions of the other regions are structurally rearranged, suggesting that HOT regions are likely to evolve in non-conserved regions between two species. As there is hardly any population genomics data available for *A. lyrata*, we cannot define HOT regions with *A. lyrata*. This analysis is now described in the main text.

2) How reliable is the LD analysis around and across HOT regions from 1,135 genomes given that the SNP data comes from short-reads? I wonder if the LD comparisons that authors used are valid

considering potential SNP acquisition bias. For example, SNP obtained within HOT regions would be less representative than the rest of the region under queries.

The regions next to the HOT regions are usually syntenic regions in which SNPs can be well accessed and also support the underlying assumption of LD analyses. As you mention, SNPs in HOT regions carry the risk to probe non-allelic regions leading to potentially mis-leading results. Being aware of this, we have taken particular care during the selection of these markers to enrich for markers that are not affected by rearrangements. However, we agree that this filtering might not find all mis-leading, non-syntenic SNPs. As a consequence of their false (noisy) alleles, such non-allelic markers would lead to a reduction in LD values and thus would not affect our conclusion that LD in HOT regions is higher as compared to the break region between HOT and syntenic regions.

Minor comments are as follow.

Line 89

Salome PA et al. 2012 (Pubmed ID 22072068) should be added. This work clearly stated the inversion on Chromosome 3 in Sha.

We agree, though would like to point out that Salome et al also state that the inversion “had been inferred before from the absence of recombination in the Bay-0 × Sha and Col-0 × Sha RILs (Loudet et al., 2002).” However, as the inversion was again confirmed by Salome et al we added this reference as well.

Line 159

HR shall be spelled out as HOT region since this is the only place where the authors used a shorten form. HR is often used to refer the hypersensitive response in plant immunity research.

Agreed and changed.

Reference 48 needs to be updated with full citation.

Thanks, corrected.

The following lines need to be checked for typos: Line 31 (at the same time), 73 (double usage of ,which), 82 (abundantly), 114 (In contrast), 249 (full).

Corrected as suggested.

Regarding Figure 4 and other supplementary Figures visualizing DM loci, authors may check if the width of NLR genic regions are properly scaled. Compared to Col-0 ones, the ones in other accessions generally look slimmer. It would be also good to state if each colored region was curated for protein coding such as early stop codon (and thus become slimmer) or merely coloring shared genic regions regardless of protein sequences.

Thank you for pointing this out. The visualization of the Col-0 gene models included 5'-UTR and 3'-UTR regions, while the UTR regions in other genomes were not shown. We have now corrected this in all relevant figures (Figure 3, 4 and Supplementary Figure 6-12).

In the methods, it should be stated whether or not the authors extracted DNA from single individuals or pooled samples. Since there are multiple versions of accessions circulating among research groups, the authors may consider stating their corresponding CS # and/or depositing their materials to stock center if single seed descendants are available.

The DNA was extracted from pooled individuals. We have added the stock center IDs to supplementary table S1, and further we updated the methods to inform exactly on the samples used.

REVIEWERS' COMMENTS:

Reviewer #1 (Remarks to the Author):

The authors have presented a lovely study. They have revised the initial submission nicely and I have no further suggestions.

Reviewer #2 (Remarks to the Author):

I have read the authors' "response to reviewers" letter and find it to be very thorough and responsive.

I have also briefly read the revised manuscript.

Although it appears that some of their responses have not been captured in the revised manuscript, they nevertheless satisfy my questions and it can be argued that they are not needed in the manuscript itself.

From my perspective, all concerns raised in my review have been answered satisfactorily.

Reviewer #3 (Remarks to the Author):

The authors sincerely addressed all the comments that I and other reviewers raised during this round of revision. The current manuscript reads well with compelling evidence. The story line improved much upon incorporating 1) comparison to Col-0 and analysis of the collapsed region; 2) clear definition on altered evolutionary dynamics; 3) cross-species comparison to highlight lineage-specific HOT events.

Again, this manuscript would need publicity in immediate future.

Reviewer #1 (Remarks to the Author):

The authors have presented a lovely study. They have revised the initial submission nicely and I have no further suggestions.

Thank you very much for your support.

Reviewer #2 (Remarks to the Author):

I have read the authors' "response to reviewers" letter and find it to be very thorough and responsive.

I have also briefly read the revised manuscript.

Although it appears that some of their responses have not been captured in the revised manuscript, they nevertheless satisfy my questions and it can be argued that they are not needed in the manuscript itself.

From my perspective, all concerns raised in my review have been answered satisfactorily.

Thanks a lot for your support.

Reviewer #3 (Remarks to the Author):

The authors sincerely addressed all the comments that I and other reviewers raised during this round of revision. The current manuscript reads well with compelling evidence. The story line improved much upon incorporating 1) comparison to Col-0 and analysis of the collapsed region; 2) clear definition on altered evolutionary dynamics; 3) cross-species comparison to highlight lineage-specific HOT events.

Again, this manuscript would need publicity in immediate future.

Many thanks for your support.